# Comparison of methods for rhythm analysis of complex animals' acoustic signals

**Lara S. Burchardt**[1,2]*, **Mirjam Knörnschild**[1,2,3]

**1** Museum für Naturkunde, Invalidenstraße, Berlin, Germany, **2** Animal Behavior Lab, Free University Berlin, Berlin, Germany, **3** Smithsonian Tropical Research Institute, Barro Colorado Island, Balboa, Ancón, Panamá

* l.s.burchardt@gmx.de

**Data Availability Statement:** All relevant data are within the manuscript and its Supporting Information files.

**Funding:** LSB received an Elsa-Neumann Stipend by the Landesgraduiertenförderung Berlin. MK was

## Abstract

Analyzing the rhythm of animals' acoustic signals is of interest to a growing number of researchers: evolutionary biologists want to disentangle how these structures evolved and what patterns can be found, and ecologists and conservation biologists aim to discriminate cryptic species on the basis of parameters of acoustic signals such as temporal structures. Temporal structures are also relevant for research on vocal production learning, a part of which is for the animal to learn a temporal structure. These structures, in other words, these rhythms, are the topic of this paper. How can they be investigated in a meaningful, comparable and universal way? Several approaches exist. Here we used five methods to compare their suitability and interpretability for different questions and datasets and test how they support the reproducibility of results and bypass biases. Three very different datasets with regards to recording situation, length and context were analyzed: two social vocalizations of Neotropical bats (multisyllabic, medium long isolation calls of *Saccopteryx bilineata*, and monosyllabic, very short isolation calls of *Carollia perspicillata*) and click trains of sperm whales, *Physeter macrocephalus*. Techniques to be compared included Fourier analysis with a newly developed goodness-of-fit value, a generate-and-test approach where data was overlaid with varying artificial beats, and the analysis of inter-onset-intervals and calculations of a normalized Pairwise Variability Index (nPVI). We discuss the advantages and disadvantages of the methods and we also show suggestions on how to best visualize rhythm analysis results. Furthermore, we developed a decision tree that will enable researchers to select a suitable and comparable method on the basis of their data.

## Author summary

In the analysis of animal communication more and more interest is shown in rhythm of animal communication and what information this might convey. In this paper, we establish a workflow to analyze the temporal structure–namely the rhythm–of any particular animals'acoustic signal with methods that are applicable for a wide range of signals and results that are easily comparable and interpretable. This workflow will enhance the understanding of rhythmicality in animals' acoustic signals as well as facilitate comparison between species. Methods we conducted ranged from simple distributional and visual

supported by a Heisenberg Fellowship (DFG KN935 3-1) from the German Research Foundation. The funders had no role in study design, data collection and analysis, decision to publish, or preparation of the manuscript.

**Competing interests:** The authors have declared that no competing interests exist.

analysis to higher mathematics such as Fourier analysis. All analyses rely on Inter-Onset-Intervals, the duration between the beginning of one element and the next. We used different datasets from two neotropical bat species as well as from the sperm whale. With this selection, we cover very short sequences with only few elements up to sequences of around 200 elements, multisyllabic and monosyllabic sequences and social communication as well as sounds used for orientation and foraging.

This is a *PLOS Computational Biology* Methods paper.

## Introduction

Rhythms can be found anywhere in the world: our hearts have rhythms, circadian rhythms are all around, music across all cultures shares certain components such as rhythm, public transportation (should) follow a certain schedule which in fact is nothing but rhythm. We learn more and more about how important a certain temporal structure is in human language, in their production as well and probably even more so in their perception; stuttering, for example, is most likely connected to a misfunction of rhythm perception [1]. This raises the question of whether rhythms, or temporal structures to use a more precise terminology, play an equally important role in animal communication and sound production. Can we learn something about rhythm in animals that will help us understand their communication better and also find underpinnings of the abundance of rhythm in human biology and culture?

Rhythm has a very narrow definition in musicality studies that does not necessarily fit the focus of this paper. We are describing temporal structures and are searching for periodicity. To prevent confusion and since terms might be used in different contexts depending on the research area, we define some key terms in a glossary (Table 1). Nevertheless, we still use the term 'rhythm' as a concept that will be understood by a broad audience, as most people have an intuitive understanding of 'rhythm', independent of whether this study analyses 'rhythm' in the musicological sense of the term.

The rhythmicality of animals' acoustic signals has an impact on a vast field of related questions. For instance, the evolution of music is investigated in the field of biomusicology, a research area that studies musicality in animals–where musicality is used as a term for different traits that occur spontaneously and are based on and constrained by biology and cognition in an animals' acoustic signals, such as harmony, timbre or rhythm [2, 3]. Moreover, knowledge about temporal structures is necessary to find coupled biological processes, such as the correlation between beat frequencies in bat's acoustic signals (also called vocalizations) with their wingbeat frequencies (i.e. wingbeats per second), independent from whether a bat might actually be flying in a vocalizing context or hanging in a roost [4]. Rhythmicality might also influence mate choice and individual recognition [5, 6]. Furthermore, neural correlates might play a role so that careful rhythm analysis can give insights into internal clocks or the importance of certain brain waves on different behavioral aspects such as the production of acoustic signals[5, 7]. Rhythm analysis can also be used to disentangle cryptic species (distinct species that are combined under one species name, because they cannot be distingzished morphologicaly) that produce sounds in different rhythms [8] or is informative in the context of vocal production learning, a part of which is for the animal to learn the correct temporal structure of a

**Table 1. Glossary.**

| Glossary | |
| --- | --- |
| Animals' acoustic signals | All acoustic signals that animals produce on purpose |
| Animal communication | The entirety of sounds and vocalizations animals produce willingly to communicate with each other |
| Vocalization | A sound produced on purpose; sound origin: vocal cords; a species can have various vocalization types |
| Social vocalization | A vocalization uttered in a social context, e.g. isolation calls |
| Isolation call | Uttered by pups to solicit maternal/paternal care |
| Animal Sounds | Willingly produced sounds by animals, with another origin than vocal cords, e.g. whales produce their sounds not with vocal cords; the term vocalizations could be misleading in this context |
| Musicality | different traits that occur spontaneously and are based on and constrained by biology and cognition in an animals' acoustic signals, such as harmony, timbre or rhythm |
| Rhythm | e.g. an ordered and recurrent alternation of different elements in a sequence of sound and silence in speech, music or animals' communication |
| Periodicity | underlying reoccurring pattern describing as sequence as periodic, e.g. an isochronous pattern |
| Isochrony | A stereotyped pattern with same beat and same gap length (gaps and beats do not have to be the same length as well, though), a metronome like acoustic pattern/beat |
| Heterochrony | A pattern with more than one underlying beat |
| Beat | The unit to describe an isochronous pattern, given in Hertz (beats per second); a beat frequency of 5 Hz would describe a sequence with an underlying pattern of 5 beats per second, i.e. 5 vocalizations per second or a temporal structure where elements are distributed regularly in a way that you could fit a maximum of 5 element into one second |
| Inter-Onset-Interval | In a sequence of acoustic signals, the time span between the start of an element and the next element, comprising the element duration and the following gap duration; in other contexts also called Inter-Pulse-Interval, Inter-Click-Interval or Inter-Call-Interval |
| Element | The smallest subunit of a sequence of acoustic signals, i.e. a distinct syllable, call, click, pulse etc. surrounded by silence |
| Exact beat frequency | The beat frequency we calculated to describe a specific sequence best (e.g. 5 Hz as in 5 beats per second) |

signal [9]. A growing body of research is addressing questions on rhythm in animal vocalizations and animal sounds (in contrast to vocalizations, sounds are produced by something other than vocal cords, e.g. sperm whale clicks; both are combined under the term acoustic signals). But before we can elaborate on this, it is important to again note different connotations of rhythm in this context. Where we speak of rhythms in animals' acoustic signals a musicologist might only talk about different beats and tempi. What we mean in this paper with rhythm and the connotation of rhythm used in other studies on the subject [4, 5, 10] describes a temporal structure that might have varying complexity but is mostly based on an isochronous beat (i.e. sounds produced by a metronome). These isochronous beats might be produced in different tempi by different species and individuals. Therefore one could also say, we search for periodicity in animals' acoustic signals. The definition for periodicity we use here is the following: we regard a sequence as periodic, when there is an underlying isochronous pattern describing it. An isochronous rhythm is a metronome like rhythm with the same beat and the same gap length' (although beat and gap length are not necessarily similar). Not every beat of that isochronous sequence needs to be corresponding with an element in the sequence that is analyzed. A beat here is every element of the isochronous pattern. It is also the actual 'beat frequency' of the isochronous rhythm. We refrain from using the word 'pulse', to prevent confusion with the use of the word 'pulse' in echolocation research. Keeping these definitions in

mind, we are still using the term "rhythm" as a summary of these conxepts in the text for reasons of readability and understanding.

Exemplary studies on the rhythmic production of acoustic signals come from male zebra finches (*Taeniopygia guttata*) [5], the bat *Saccopteryx bilineata* [4] or the humpback whale (*Megaptera novaeangliae*) [11]. While male zebra finches sing with different rhythms depending on the individual, *S. bilineata* vocalizations share a common temporal structure, likely coupled to wingbeat frequencies [4, 5]. Yet another pattern was found in the song of humpback whales (*Megaptera novaeangliae*), where individuals can produce very stable temporal structures or sound sequences that vary rapidly in tempo and rhythm [11].

Other forms of rhythm production were found in the palm cockatoo (*Probosciger aterrimus*). The males of this species drum quasi-isochronous patterns, using tools, in a consistent manner [12]. Chimpanzees use individual rhythm signatures—likely in a fashion to help recognize unseen companions–when cracking baobab fruits [13].

Studies on the perception of rhythms or periodocity deal for example with the ability of animals to discriminate rhythms, e.g. in rats and European starlings [14, 15]. Moreover, the first instance for a biologically relevant rhythm in non-human mammalian acoustic signals was found in the northern elephant seal, where males can discriminate between familiar and unfamiliar male opponents using the temporal structure of vocalizations. Rhythms apparently differ between individuals in a way that facilitates the discrimination of individuals [6].

With the growing body of studies and its implications for other research questions, it is important to present methods in a reproducible way and find methods that are applicable to a vast majority of datasets in which temporal structures can be analyzed. Reproducibility, interpretation biases, p-hacking (the distortion or manipulation of results through data mining) and apophenia (the tendency to see a pattern in random data) are key issues in all research fields. Defining clear methodologies with open access to code and data is one way of tackling those issues [16]. Results must be clearly structured and comparable between species and contexts. A number of papers address these issues and describe suitable methods by means of artificial data, with a decision tree depending on the respective question [10]. Nevertheless, a comparison of different methods on different original datasets and of the influence of differences in datasets on the decision for a method is missing, even though this would help researchers to choose which methods to use for their data depending on the question at hand.

Acoustic recordings can differ enormously in their features. Depending on the recording situation and signals to record, one faces very different sampling rates and recording lengths. Moreover, the number of elements (i.e. a distinct syllable, call or click in a given sequence, surrounded by silence) in a recording differs greatly as well as element durations, noise level or amplitudes. Also, the recording situation differs a lot between a zebra finch recorded in a controlled recording box, a whale tagged with a recording device in the Pacific Ocean or a bat vocalizing in its roost. Nevertheless, all these acoustic signals are suitable and interesting to check for periodicity (or rhythmicality). It is crucial that a comparable method can be applied to all these different recordings. Methods that have been used for rhythm analysis include Fourier analysis [4, 5, 10, 17] or calculation of nPVIs, the normalized Pairwise Variability Index, which was originally developed to assess temporal variability in human speech rhythm [10, 18–21]. The nPVI is a measure of variability between Inter-Onset-Intervals. It will be zero for a perfectly isochronous sequence with all Inter-Onset-Intervals being equal. Furthermore different variations of the analysis of Inter-Onset-Interval–the duration between two adjacent elements (IOI [11, 22, 23]; also called Inter-Pulse-Interval, IPI [24] or Inter-Click-Interval, ICI [25–27])—and a so called generate-and-test or GAT approach [4, 5, 10] were used in rhythm analysis so far. All these methods search for isochronous patterns, therefore, again, we are rather searching for periodicity and isochronous beats underlying a sequence.

This paper aims to help researches decide on a method for the analysis of the temporal structure of their biological data. Five methods were used on three different datasets to assess 1) what kind of rhythm an acoustic signals might have (e.g. isochronous vs. heterochronous) and 2) which exact beat frequencies describe a given sequence best. Rhythm analysis can be done on different levels (Fig 1). Depending on the question at hand and the detail of the analysis, different methods can be used. At first, one has to establish whether a given acoustic signal sequence is rhythmic (periodic) at all. The general hypothesis is that a signal is periodic. This can be assessed by a detailed analysis of Inter-Onset-Intervals (IOIs) and by visual assessment of the data. The next step is to decide whether a signal shows an isochronous–that is a metronome-like–rhythm or a heterochronous rhythm. This again can be inferred from IOI analysis and nPVI calculations. If an isochronous rhythm is to be detected and one wants to know the exact beat frequencies of a signal, a Generate-and-test approach (GAT) [4, 5, 10] or a fast Fourier transformations (FFT [4, 5, 10, 17]) can be used; which one to use depends on the data. We developed a goodness-of-fit value for exact beat frequencies calculated with FFT and by IOI analyis, as these were missing so far. This makes it now possible to not only infer exact beat frequencies but how good a beat frequency actually fits a dataset and how good a 'beat producer' an animal is. To find an underlying pattern within or between individuals a cluster analysis can be run. If a heterochronous beat is to be expected, recurrence plots are a good way to visualize the data, to find underlying structures and to be able to decide how to proceed in the analysis. Visualizing underlying or sub-structures can also be relevant in the context of nested signals, where a small part of a sequence might have a very different tempo than the rest. In that case it might be worthwhile to rerun parts of the analysis on that specific part.We also introduce recurrence plots on isochronous data in this paper. All of the above mentioned methods were used on three datasets to compare results and to show the advantages and disadvantages of the different methods as well as their interpretation.

## Methods

### Labeling of elements and datasets

We chose three different datasets for the analysis with very different properties: 1) monosyllabic (i.e. only one element type in a sequence), short isolation calls of the neotropical bat *Carollia perspicillata*, 2) multisyllabic, medium long isolation calls of the neotropical bat *Saccopteryx bilineata*–both social vocalizations–and 3) monosyllabic, very long echolocation click trains of the sperm whale *Physeter macrocephalus* used for orientation and foraging (Fig 2). With this, we cover a broad range of possible acoustic signal sequence structures and can infer the applicability of the methods for a broad range of acoustic signals.

The basis for all analyses were element onsets. An element is a distinct syllable, call or click in a given sequence that is surrounded by silence. It is necessary that elements and their onsets are clearly recognizable. For each acoustic signal sequence, the on- and offset of its elements were determined for subsequent analyses. For multisyllabic isolation calls of *S. bilineata*, we manually determined element on- and offsets based on oscillograms (see [29] for details). For sperm whale echolocation click sequences and isolation call bouts of *C. perspicillata*, we used an automatized procedure in Avisoft SASLab Pro (based on amplitude detection threshold; -20 dB relative to the element's peak frequency for bats; adjusted manually to not include buzzes for sperm whales) to determine element on- and offsets.

We analyzed multisyllabic isolation calls from 5 pups of *S. bilineata* (see [29] for details on study site and sound recordings). Each isolation call contained 5–26 elements, i.e. syllables (14 ± 3.5, mean ± SD) and was composed of 2–4 different element types (mean: 3 element types), but this distinction was not relevant for further analyses. Furthermore, isolation call

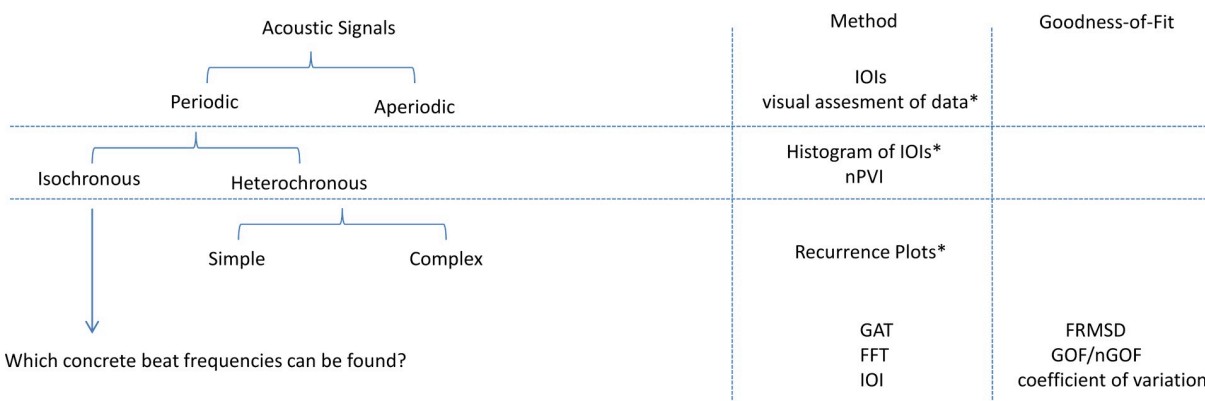

**Fig 1. Which methods to use depending on the level of analysis: A first evaluation of whether a signal is periodic or aperiodic relies on IOI and visual assessment of the data.** Whether an acoustic signal sequence might be isochronous or heterochronous can be inferred from IOIs and nPVI calculations. To find exact beat frequencies a GAT approach, FFTs or again an assessment of IOIs can be used, and the detection of simple or complex heterochronous patterns is guided visually by recurrence plots. Exact beat frequencies are only interpretable if accompanied by a goodness-of-fit value. The figure was adjusted after [28].

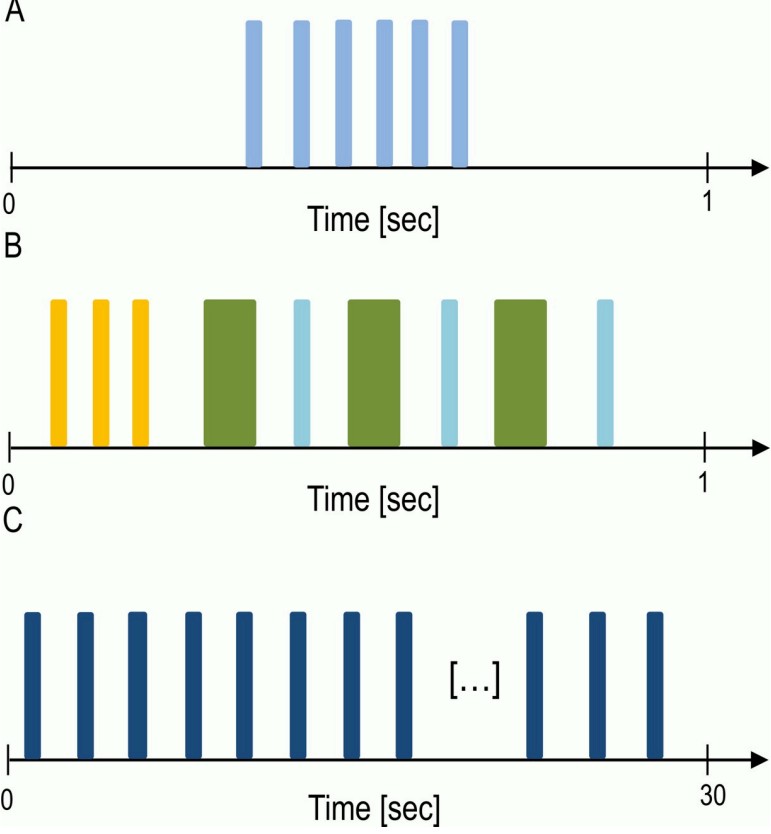

**Fig 2. Visual representation of the different sequences.** Different colors indicate different element types. (A) An exemplary sequence of *C. perspicillata* isolation calls. (B) An exemplary sequence of *S. bilineata* isolation calls. (C) An exemplary sequence of *P. macrocephalus* echolocation clicks as used for orientation and foraging. Click trains can be up to 200 elements long.

bouts of 5 *C. perspicillata* pups were analyzed (see [30] for details on study site and sound recordings). Each bout contained 3–11 elements (mean: 3 elements) and was composed of a single element type. We assessed a total of 47 bouts (Pup 1: 11 bouts, Pup 2: 8 bouts, Pup 3 and 5: 9 bouts, Pup 4: 10 bouts). Furthermore, we analysed 60 sequences of echolocation clicks from a single deep dive of the female sperm whale Sophocles, recorded by the Dominican sperm whale project on 24. April 2014 (for details on study site and recordings see [31, 32]). We extracted trains manually with the software CoolEdit 2000. Single trains were distinguished visually by a clear silent gap of at least 3 seconds (in most cases at least 5 seconds). The elements were afterwards labeled with the software Avisoft SASLab Pro; only the search phase was labelled and feeding buzzes–if at all present–ignored. Feeding buzzes can occur at the very end of a click train when an animal is hunting; they are characterized by a higher repetition rate and less energy [33]. Trains contained 13 to 248 elements, i.e. clicks (115 ± 48, mean ± SD).

### Rhythm analyses

The different methods used are IOI analyses, including the calculations of coeffiecients of variation and three methods using the IOIs as input, namely nPVI calculations, Fourier analyses, and a generate-and-test approach. IOIs can be used to visualize the data in histograms or recurrence plots. When one wants to find the exact beat frequencies which best describe an acoustic signal sequence Fourier analysis, IOI analsyis, and the GAT approach can be used. To assess how good any of those exact beats describe a given sequence, goodness-of-fit values are crucial. Different values serve as a proxy for the goodness-of-fit of the best fitting beat in the three different methods and play an important part in the interpretability and comparability of results between species and studies.

**IOI.** The Inter-Onset-Intervals (IOI) were assessed and the mean IOI of each sequence converted into the corresponding exact beat frequency by dividing it by 1 [as Hertz is 1/second]. The coefficient of variation was calculated as an indicator of variability. It is estimated as the ratio of the standard deviation to the mean of the sample ([34], Eq 1). The formula for an unbiased estimator ([35], Eq 2) was used.

$$\widehat{C_V} = \frac{s}{\bar{x}} \tag{1}$$

$$\widehat{C_V^*} = \left(1 + \frac{1}{4n}\right)\widehat{C_V} \tag{2}$$

$\widehat{C_V}$ = *Coefficient of Variation*   $\widehat{C_V^*}$ = *unbiased Coefficient of Variation*

$s$ = *standard deviation*   $\bar{x}$ = *sample mean*   $n$ = *sample size*

**nPVI.** Two adjacent IOIs were compared: their difference was calculated and divided by their average; the nPVI gives the average of all these ratios in a sequence multiplied by 100. The obtained values have little explanatory power, beyond being able to assess whether a sequence is isochronous or not [10, 18, 19, 21]. We calculated nPVI for all sequences of a dataset separately (named 'sequence' in results) and for all IOIs of a dataset combined (named

'overall' in the results).

$$nPVI = \sum_{k=1}^{m-1} \left| \frac{IOI_k - IOI_{k+1}}{\frac{IOI_k + IOI_{k+1}}{2}} \right| * \frac{100}{m-1} \tag{3}$$

$nPVI = normalized\ Pairwise\ Variability\ Index \quad k = index\ number$

$m = total\ number\ of\ indices \quad IOI = Inter - Onset - Interval$

**Recurrence plots.** In a recurrence plot higher-order patterns within an acoustic signal sequence can be visualized. It plots the sequence of IOIs as their differences, building a raster showing the differences between every IOI with every n-th IOI. The differences are marked by color code (for code see [10]). Both axes represent the IOI indices in their sequential order.

**Fourier analysis.** Timestamps of element onsets were used to form a binary point process. Sequences with a time resolution of 5 ms were created, in which only events (i.e. element onsets) were represented by '1', everything else in the ssequence was represented by '0'. Each sequence started and ended with an event, represented as a '1'. A fast Fourier transformation was calculated (FFT). After that, frequencies of maximum power were selected as 'best fitting beat' [4, 17], which are the exact beat frequencies we subsequently described a sequence with.

A normalized goodness-of-fit value based on the zero-bin component (DC Offset) of the FFT signal was established. In a normal oscillating signal the zero-bin-component–the amplitude of the signal at 0 Hz–is zero. In a binary sequence, the zero-bin component is not 0 but, instead, same as the mean of the signal in the time domain (adjusted after [36]; Eqs 4 & 5); therefore it is dependent on the total number of elements and the number of samples. It thus functions as an internal reference (Fig 3).

$$X(f) = \frac{1}{N} \sum_{n=0}^{N-1} x(n) e^{-j2\pi \frac{f}{N}n} \tag{4}$$

$$X(0) = \frac{1}{N} \sum_{n=0}^{N-1} x(n) \tag{5}$$

$X(f) = Signal\ frequency\ domain \quad N = sample\ size \quad j = j\ function \quad e = Euler\ number$

$n = index\ number \quad x(n) = signal\ in\ time\ domain \quad f = frequency$

The nGOF value is calculated by dividing the amplitude P of the best fitting beat frequency ($P_{best}$) by the amplitude P of the zero-bin-component ($P_0$) multiplied with the sampling length (L) (Eq 6).

$$nGOF = \frac{|P_{best}|}{L * |P_0|} \tag{6}$$

$nGOF = normalized\ Goodness\ of\ Fit\ value \quad P_{best} = highest\ Amplitude$

$L = sampling\ length \quad P_0 = Amplitude\ at\ 0\ Hz, zero - bin - component$

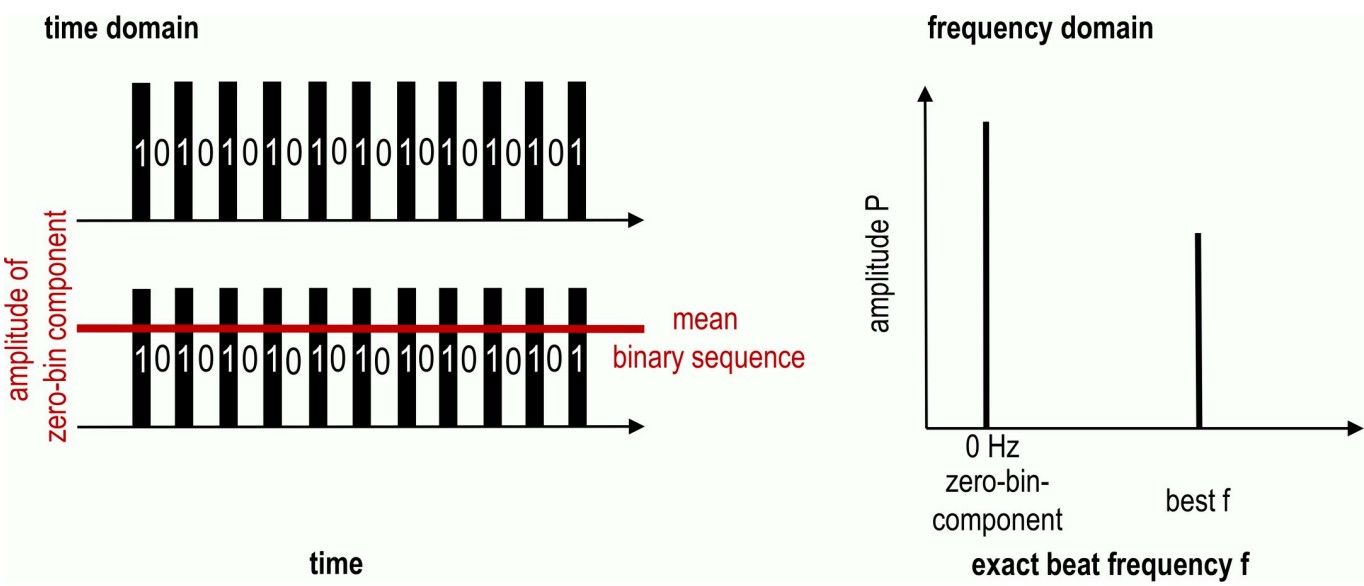

**Fig 3. Visual explanation of the internal reference: The mean of the binary sequence that serves as input for the Fourier analysis determines the amplitude of the zero-bin-component (DC-term).** This amplitude will always be the highest in this kind of analysis serving as an internal reference for the second highest peak that determines the best fitting exact beat frequency.

**GAT.** In the generate-and-test approach (developed by [5]) the original sequence of element onsets gets tested against computed perfectly isochronous onset sequences of a predefined frequency window (i.e. 5–100 Hz as in beats per second). Sequences were computed in a frequency window from 2–100 Hz in 0.01 Hz increments. For each beat frequency, the the root-mean-square deviation (RMSD) of all elements in a sequence from their nearest single beat was calculated. The parameter was then normalized for frequency (by dividing it by the frequency), resulting in the frequency normalized root-mean-square deviation–FRMSD.

**Artificial data.** To further the understanding of the analysis principles we ran all methods on three artificial datasets: 1) perfectly isochronous sequences with IOIs of 0.1, 0.3 or 0.5 seconds; 2) Ten sequences á 100 elements randomly drawn from a uniform distribution between 0 and 1 and 3) three subdatasets, that were drawn from a Gaussian distribution with means of 1, 0.2 or 0.1 seconds with standard deviations of 0.5, 0.1 and 0.05 respectively. Again each data set consisted of 10 sequences with 100 elements in each sequence. Negative numbers were permitted and the drawing of a negative number re-run until a positive number was selected. This was done manually (dataset 1) and in Matlab with the 'rand' (dataset 2) and 'normrand' function (dataset 3).

## Cluster analysis

An agglomerative, hierarchical clustering algorithm that used the group average of frequency distances as the basis for finding clusters was applied. Dissimilarities were given by Euclidean distances; the dissimilarity threshold to find clusters was set to 0.05 for all data sets. Cluster analyses were performed for all three methods yielding exact beat frequencies in Matlab.

## Software and code

We used Matlab (Version 2017b & 2016b) and R (Version 3.5.3) for the analyses. CoolEdit 2000 (Syntrillium, Phoenix, USA) was used to extract single echolocation click trains from the dive of a sperm whale. Furthermore, we used Avisoft SASLab Pro Version 5.2.10 (Berlin,

**Table 2. Summary of IOI results.**

|  | Mean IOI [sec] | SD (σ) | Coefficient of Variation (overall) | Coefficient of Variation (mean of sequences) | Range[sec] | n IOIs | nPVI | n sequences |
|---|---|---|---|---|---|---|---|---|
| *C. perspicillata* | 0.043 | 0.013 | 0.31 | 0.23 | 0.01 – 0.1 | 195 | 2.3 to 110.7 mean 35.9 | 47 |
| *S. bilineata* | 0.078 | 0.022 | 0.29 | 0.19 | 0.028 – 0.28 | 646 | 6.4 to 99.4 mean 22.8 | 50 |
| *P. macrocephalus* | 0.46 | 0.1 | 0.22 | 0.14 | 0.03 – 3.1 | 6913 | 0.4 to 13.6 mean 5.2 | 60 |

Germany) to visualize recordings and to determine element onsets automatically (for isolation call bouts of *C. perspicillata* and click trains of *P. macrocephalus*) and manually (for multisyllabic isolation calls of *S. bilineata*).

The code for the GAT approach was published elsewhere (see [10]) and the code to run the FFT as well as exemplary data is provided here: https://github.com/LSBurchardt/FFT-Method.

## Results and methods discussion

### IOIs and nPVIs

We show key data for all datasets in Table 2: the mean of the IOIs in seconds, the standard deviation of IOIs, as well as the coefficient of variation over all IOIs of a dataset ($C_V$ overall) and the average coefficient of variation between sequences ($C_V$ sequences) of a dataset. In contrast to the commonly used parameters variance and standard deviation, the coefficient of variation is neither sample-size nor mean dependent. Therefore it yields comparable results independent of the dataset. To ensure comparability we used the formula for an unbiased estimator ([35], Eq 2) since especially for smaller sample sizes the normal coefficient of variation (Eq 1) tends to underestimate the variation.

Furthermore, we give information on the range of IOIs in seconds and the number of IOIs comprising the datasets. The range of calculated nPVIs as well as their mean is given together with the information on the number of sequences underlying the nPVI analysis and subsequent analysis of exact beat frequencies per sequence via GAT, Fourier analysis and IOI calculation.

A visual inspection of IOIs is the first step in determining the temporal structure of any given dataset. A unimodal distribution of IOIs is a strong indicator for isochrony because all IOIs spread around the one most prominent duration category. The steeper the distribution, the more consistent an isochronous pattern should be. We find unimodal distributions for all three datasets (Fig 4, first column). To quantify the temporal structure, we then look at different key data values: first the $C_V$ (sequences) and the nPVI.

The smaller the $C_V$ (sequences), the less variation we find in IOIs of a dataset, indicating a more consistent structure and possibly isochrony. Smaller nPVIs suggest a similar interpretation. A small nPVI value does not only show a consistent structure but an isochronous structure. When interpreting nPVI values we must consider that even though a very small nPVI indicates isochrony, a middle (20–40) or even high (60–100) nPVI does not necessarily disagree with isochrony and definitely not with rhythmicity. In a computer simulated element sequence with a stress pattern, namely a pattern with an isochronous occurrence of stressed elements, an nPVI value of 94.54 was calculated (see [10]). An indicator of variation between sequences and possibly between individuals is the difference between the $C_V$ (sequences) and $C_V$ (overall). The $C_V$ (sequences) should always be smaller than the $C_V$ (overall), the bigger the

difference between the two, the higher the variation between sequences and possibly individuals (S1 Table for examples on artificial data).

Looking at the results for our datasets, we can infer isochrony for all three datasets, with *P. macrocephalus* showing the strongest patterning and likely a very strict isochrony and only a few variations between sequences. *S. bilineata* and *C. perspicillata* show values that hint at an underlying isochronous structure with small (*S. bilineata*) and medium (*C. perspicillata*) differences between sequences and individuals.

## Exact beat frequencies

After the overall analysis of the pattern, it is interesting to analyze exact beat frequencies, which would describe individual sequences best. Depending on the results of the overall patterns (isochrony or not, high variability vs. low variability), different methods are appropriate to analyse these exact beat frequencies. For example, if results indicate a higher probability of differences between sequences and individuals, an IOI analysis would oversimplify results and we do not consider it fitting. In that case GAT analysis is useful. Nevertheless, if the overall pattern suggests a very strong rhythm, the computationally more intensive analysis of the GAT

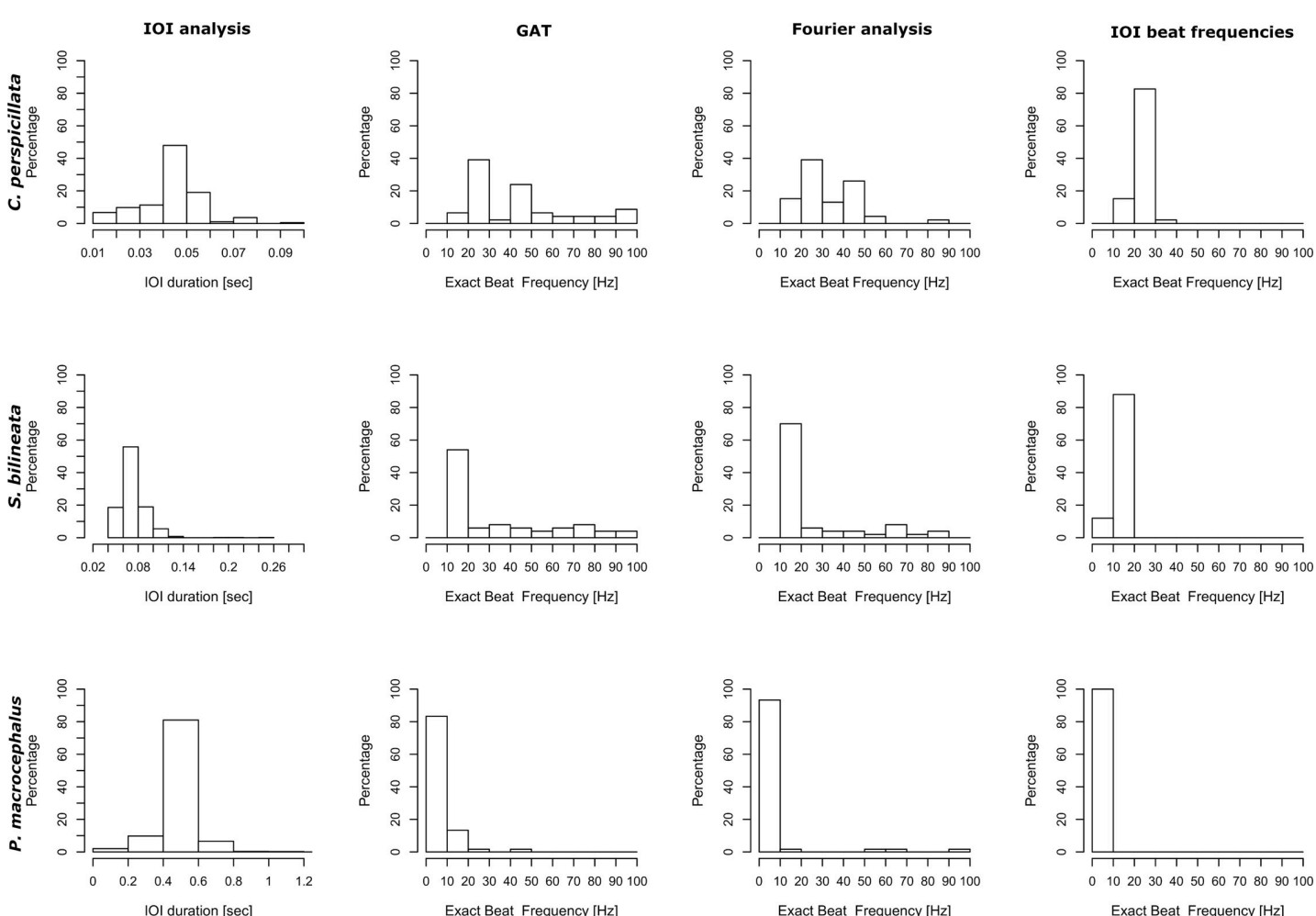

**Fig 4. Analysis of the datasets per method: The first column shows the distribution of IOIs for all datasets.** The second to fourth column depict exact beat frequency distributions for the three datasets (1. *C. perspicillata*, 2. *S. bilineata*, 3. *P. macrocephalus*) and different methods.

approach can be spared, because it would most probably not add substantially to the results of the IOI analysis or Fourier analysis. Fourier analysis is a very strong tool to analyse rhythm, but also needs some consideration, for example when deciding which time resolution to choose for the binary sequence. The rather coarse time resolution of 5 ms used in our analysis was chosen for different reasons. A time resolution of 5 ms results in a sampling rate of 200 Hz. Since in an Fourier analysis, signals up to half of the sampling rate can be deconstructed, a sampling rate of 200 Hz will result in frequencies between 0 and 100 Hz being analyzed. In other studies, 100 Hz as the upper boundary for the investigation proved suitable for bird song as well as the much faster echolocation pulses of neotropical bats [4, 5]; therefore, we also used this frequency window for the analysis here. Another very important point to be kept in mind: the chosen time resolution directly influences the frequency resolution of the Fourier signal; the higher the time resolution, the lower the frequency resolution will be and vice versa (Eq 7). This problem diminishes with long sampling length but especially in short signals of under and around 1 second, it is a considerable issue. Our chosen time resolution gives suitable frequency resolutions even with short sampling length.

$$\frac{temporal\ resolution}{sample\ length} = frequency\ resolution \tag{7}$$

Keeping advantages and disadvantages in mind, one should always run more than one analysis method to get a better picture of the data at hand. Our results for the exact beat frequencies are presented in Table 3. For each method for calculating exact beat frequencies (GAT, FFT, IOI) the range of detected beat frequencies is given for the three datasets. The results cluster around certain values. We divided the frequency window we looked at (0–100 Hz) in 10 Hz categories; one category will encompass most of the found sequences (i.e. the category 20–30 Hz). This most prominent category is given alongside the percentage of sequences showing beat frequencies in that category. In addition, the results are visualized in Fig 4, with the different methods in columns and the datasets as rows. The most prominent categories are clearly visible in the histograms for all methods and datasets.

## Goodness-of-Fit

Finding such strong categories as we can see in the histograms (Fig 4) hints at an underlying isochronous pattern and we can be sure that the exact beat frequencies we found describe the sequences well. It is very unlikely that we find random exact beat frequencies by chance that show such a pattern of up to 100% of beats found falling into the same bin category. But what if such an overall pattern is uniformly distributed? How can we be sure that we did not find random beats and how can we compare species and contexts with regards to how well a single beat describes a sequence? For that, we used and developed different goodness-of-fit values which quantify how well a beat describes a sequence.

There are different ways to assess the goodness-of-fit of a beat. By and large, it represents how close the original sequence of elements is described by one certain beat. Since we are searching for the best fitting beat, it describes how well this beat describes the sequence. The goodness-of-fit values for the different methods are correlated to different measures like the number of elements and length of the sequence, and sometimes to a certain extent to beat frequencies; they fall on very different scales and therefore need careful consideration (S3 Table for examples on artificial data).

For the GAT approach, the FRMSD (Frequency-normalized Root Mean Square Deviation) depicts the goodness-of-fit. It is positively correlated to the number of elements in a sequence in a non-linear way, but superior to the RMSD which is in addition highly frequency

**Table 3.  Overview of exact beat frequencies found for three datasets with three methods.**

| | GAT | | | FFT | | | IOI | | |
|---|---|---|---|---|---|---|---|---|---|
| | Min [Hz] | Max [Hz] | Prominent category [Hz] and % | Min [Hz] | Max [Hz] | Prominent category [Hz] and % | Min [Hz] | Max [Hz] | Prominent category [Hz] and % |
| *C. perspicillata* | 17.9 | 100 | 20–30 39.1% | 11.8 | 83.3 | 20–30 39.1% | 12.4 | 30.6 | 20–30 82.6% |
| *S. bilineata* | 8 | 100 | 10–20 54% | 11.4 | 86.6 | 10–20 70% | 8.1 | 17.1 | 10–20 88% |
| *P. macrocephalus* | 2 | 40.9 | 0–10 83% | 1.7 | 93.7 | 0–10 93.3% | 1.9 | 2.4 | 0–10 100% |

dependent. Using the FRMSD results in finding the slowest beat, coinciding best with element onsets [4, 5, 10]. It describes the average temporal deviation as a fraction of a full cycle and therefore has no unit [4, 5]. For the most part, FRMSD values for *C. perspicillata* pups overlap with FRMSD values in *S. bilineata* pups. Nevertheless, the minimum value we find in *S. bilineata* is much higher, while the highest value is lower than in *C. perspicillata* pups. Goodness-of-fit values for the GAT approach show a much broader range for *C. perspicillata*. Element numbers in *S. bilineata* pups are 2- to 9-fold higher, therefore values for *S. bilineata* are considered to show a better fit than the ones for *C. perspicillata*. Due to the FRMSDs positive correlation to element numbers, it is not surprising that we find higher values in the very long sequences of *P. macrocephalus*. Exact values for all three species are shown in Table 4.

For the Fourier analysis the basis for the goodness-of-fit is the amplitude of the Fourier signal. The amplitude P of the Fourier signal, which is used to determine the best fitting beat, is also indicative of how good the beat actually fits: the higher the amplitude, the better the fit. Nevertheless, the amplitude is strongly correlated to sample length and number of events in the sequence. Therefore, amplitudes could so far only be compared within one dataset and with good knowledge about the correlations. The nGOF on the other hand shows a much smaller correlation with sample length and number of events (S1 Table) and is therefore more appropriate to use as a goodness-of-fit value. The nGOF was validated by correlating it to the already established goodness-of-fit value of the Generate-and-test approach, the FRMSD value (S1 Table). The nGOF values range from 8e-6 to 1.3e-3 with a median of 2.5e-5 for *P. macrocephalus*. The measure only set into relation with the internal references–but not normalized for the length of the signal–lie between 0.22 and 0.67 (GOF) which can be thought of as the percentage this one particular beat frequency has on describing the original sequence. This value is easier to interpret, but–again–the signal length has a strong impact, which gets clear when comparing the values of the very long sperm whale click trains with the way shorter values for isolation call bouts of *C. perspicillata*, that show much higher and therefore actually "better" values. All other results, on the other hand, have to lead to the interpretation, that the sperm whale echolocation click trains are a lot more regular and therefore closer to a "perfect" beat than the bat isolation calls. This also shows in the nGOF values for the FFT of *C. perspicillata*

**Table 4.  Comparison of Goodness-of-Fit values for all datasets and methods.**

| | GAT | FFT | | IOI | |
|---|---|---|---|---|---|
| Dataset/Method | FRMSD | GOF | nGOF | $C_V$ (overall) | $C_V$ (sequence) |
| *C. perspicillata* | 0.007–0.214 | 0.57–0.98 | 0.012–0.064 | 0.31 | 0.23 |
| *S. bilineata* | 0.059–0.183 | 0.5–0.92 | 5.5e-4–0.014 | 0.29 | 0.19 |
| *P. macrocephalus* | 0.07–0.26 | 0.23–0.87 | 1.2e-5–0.0032 | 0.22 | 0.14 |

pups. They show values that are more than a thousand fold larger than in the sperm whale data.

The goodness-of-fit values for FFT analysis of *S. bilineata* isolation calls fall in between sperm whales and *C. perspicillata* pups, being 10 fold smaller than the values from *C. perspicillata* and 100 fold larger than *P. macrocephalus*. Again exact values are shown in Table 4.

In IOI analysis the sample size independent measure of the coefficient of variation ($C_V$) can be used as an indicator of the goodness-of-fit; the smaller the $C_V$, the less spread there is in the IOIs, which means they are more similar to each other, thus corresponding to a more regular beat. Since the IOI analysis bears little sequence information it is just indicative of the overall regularity. All measures are shown in Table 4. The differences between $C_V$ (overall) and $C_V$ (sequence) moreover give insight into the likelihood of finding individual differences. While in the $C_V$ (overall) all IOIs of an acoustic signal sequence are regarded, in $C_V$ (sequence) only one sequence is regarded and the average for all analysed sequences calculated. Therefore we might have individually very isochronous sequences leading to small values for $C_V$ (sequence) but very different sequences, leading to a high value for $C_V$ (overall). Therefore, the bigger the difference between $C_V$ (overall) and $C_V$ (sequence), the higher the likelihood of finding differences between individuals. The difference between the two is the smallest for *P. macrocephalus* and highest for *C. perspicillata*. This leads to the interpretation that it is most likely to find individual differences in exact beat frequency patterns in *C. perspicillata* and we do not expect them in *P. macrocephalus*.

## Cluster analysis

Visual inspection of the detected exact beat frequencies per individual confirms what the overall pattern and $C_V$ calculations already indicated. We find a pattern within individuals, where beat frequencies cluster around certain values. Depending on the method and the dataset, these clusters are differently strong and fall around different values.

The cluster analysis is a good way of depicting "preferences" of the different individuals for certain beats. In *S. bilineata* pups, clusters do not differ much between individuals and show cluster strengths of between 30 and 70% for the GAT approach; clustering the results of the FFT analysis leads to clusters containing 40% to 60% of sequences per individual. In IOI analysis, clusters contain between 60% and 100% of sequences. All strongest clusters fall between 10 and 20 Hz (also see [4]). The picture for *C. perspicillata* pups looks slightly different though. We find the strongest clusters containing a third up to 100% of sequences of an individual falling into one cluster with IOI analysis. The difference is that not all clusters lie in the same beat category. We find most of the strongest clusters between 20 Hz and 30 Hz for GAT and FFT analysis as well as between 40 Hz and 50 Hz. Other clusters fall in different categories. For IOI analysis, on the other hand, all clusters fall at least partially between 20 Hz and 30 Hz (Fig 5 and S2 Fig).

Since we analyzed echolocation click trains of a single individual for *P. macrocephalus* such a cluster analysis is not useful here. But the very strong patterning and previous research [37] let us assume that there are no significant individual differences.

## Recurrence plots

In the following section, we describe two exemplary recurrence plots, one showing a multisyllabic isolation call of *S. bilineata* and the other one showing an echolocation sequence of *P. macrocephalus*. Recurrence plots offer a visual representation of the temporal pattern of a sequence. The more uniform the sequence, the more white and light grey colors can be seen in the plot: white stands for no to very little differences between two adjacent IOIs and the darker

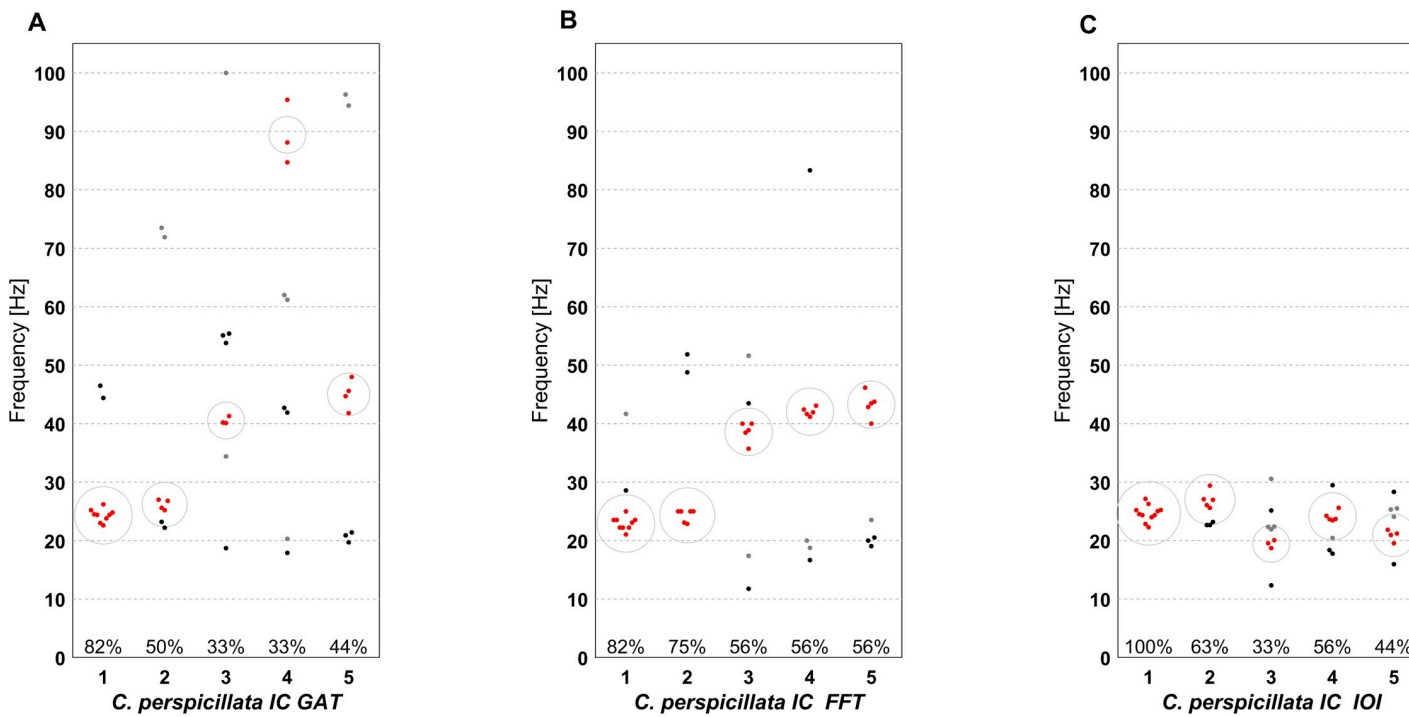

**Fig 5. Individual Beat Clusters in *C. perspicillata* pups confirm the results of other methods: Exact beat frequencies as analyzed with the three different methods are shown with clusters in the data.** One individual is depicted per column, all exact beat frequencies found are shown as dots. Depicted in red are the sequences falling into the largest cluster of sequences sharing a similar beat. Percentages at the bottom indicate the percentage of sequences per individual in the largest cluster. (a) Exact beat frequencies and individual clusters as obtained by the GAT approach. (b) Exact beat frequencies and individual clusters as obtained by the FFT method. (c) Exact beat frequencies and individual clusters as obtained by IOI analysis.

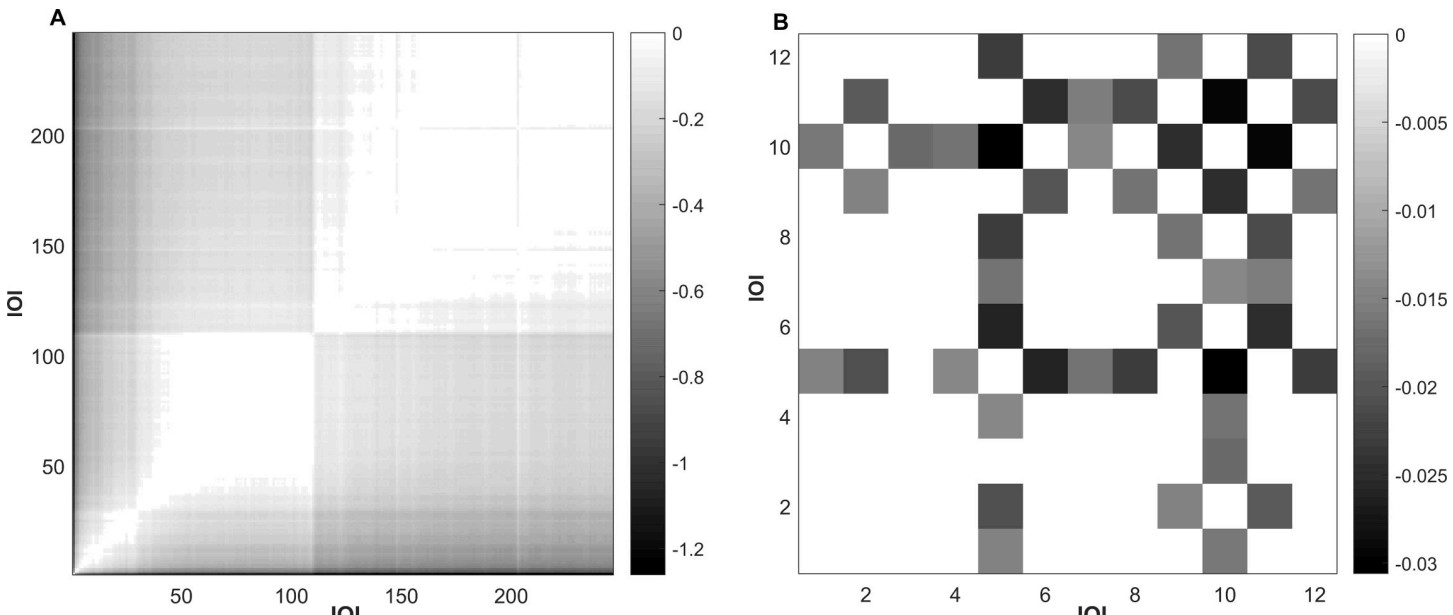

**Fig 6. Recurrence plots of two sequences.** No difference is indicated by white; the darker the color, the bigger the difference. Note that absolute differences are depicted and colors represent different differences in both plots, as shown in the legend. A) Echolocation click train of *P. macrocephalus*: a very isochronous pattern is visible by only white and light grey colors. B) The multisyllabic structure of an isolation call of *S. bilineata* is visible in the differences in IOIs: a subsequence of very similar IOIs is followed by an alternating sequence of two more element types.

a comparison, the bigger the difference. The very strict isochronous pattern of the sperm whale echolocation sequences is depicted in an almost white plot (Fig 6A). In contrast, we can even see the structure of the multisyllabic isolation call of *S. bilineata* pups in the corresponding recurrence plot (Fig 6B), where very similar IOIs are followed by slight pairwise changes of IOIs at the end of the sequence, which corresponds to changes between two element types. These plots could be a very valuable addition in the analysis of more complex temporal structures in acoustic signals because higher order structures–for example, different parts of temporal structure within one acoustic signal sequence–can be visualized and used to determine how to proceed. For very short sequences such as for *C. perspicillata* isolation calls, plotting a recurrence plot most often does not offer additional valuable insights. They are to be interpreted carefully, especially when sequences to be compared via a recurrence plot vary widely in IOI length. The same absolute difference between IOIs might be irrelevant for one but important for another species. The same color might not stand for the same absolute difference in two plots.

### Decision tree

Incorporating the different methods into a workflow that includes both the data structure as well as results of early analysis steps leads to a decision tree, describing which methods to use in what case (Fig 7).

## Discussion

This study presents a comprehensive overview of the analysis of periodicity and rhythmicality in animal acoustic signals by comparing different methods for three different original datasets and introduces two new goodness-of-fit values for rhythm analysis methods. How to decide on the fitting methods depending on the data is depicted in Fig 7.

Periodicity can be inferred for all three datasets from the results of the IOI analysis and visual assessment of the sequences: multisyllabic isolation calls of *S. bilineata*, isolation call bouts of *C. perspicillata* and echolocation click trains of *P. macrocephalus*. This information might be useful to answer a broad range of questions, but independent of the question at hand are the methods. Those methods enable us to actually infer or exclude periodicity for a given sequence. These methods are the topic of this paper.

The methods (nPVI calculations, $C_V$, IOI analyses, GAT, Fourier analyses) were adjusted by using three very different kinds of vocalization and sounds for them to be applicable to a broad range of acoustic signals. We used long and short signals in terms of overall duration and element duration, multisyllabic and monosyllabic sequences, and echolocation sequences for navigation as well as social calls. Furthermore, this ensures comparable results and fast and relatively easy implementation of the different analyses, which was the main aim of this study. Nevertheless there might be extreme examples of acoustic signals where the method (i.e. Fourier analysis' time resolution or the frequency window in the GAT approach) could need adjustments; these could include the very slow and long rumbles of elephants [38, 39] or the extremely fast and short echolocation signals of some bats such as *Kerivoula pellucida*, a small Verspertilionidae bat from Southeast Asia with element lengths of ~1.9 ms and IOIs of around 5 ms [40] or the even shorter but a little slower calls of *Micronycteris microtis* with an element length of 0.2 ms and IOIs of 14 to 30 ms [41]. For the very short elements of some bat species, the sampling rates for creating the binary sequence, serving as input for Fourier analysis, would need to be much higher for two reasons: first, with a time resolution of 5 ms and element lengths of 2 ms or even 0.2 ms, the accuracy of labelling becomes to coarse. Second, the range of frequencies a sequence is described with in a Fourier analysis is dependent on the

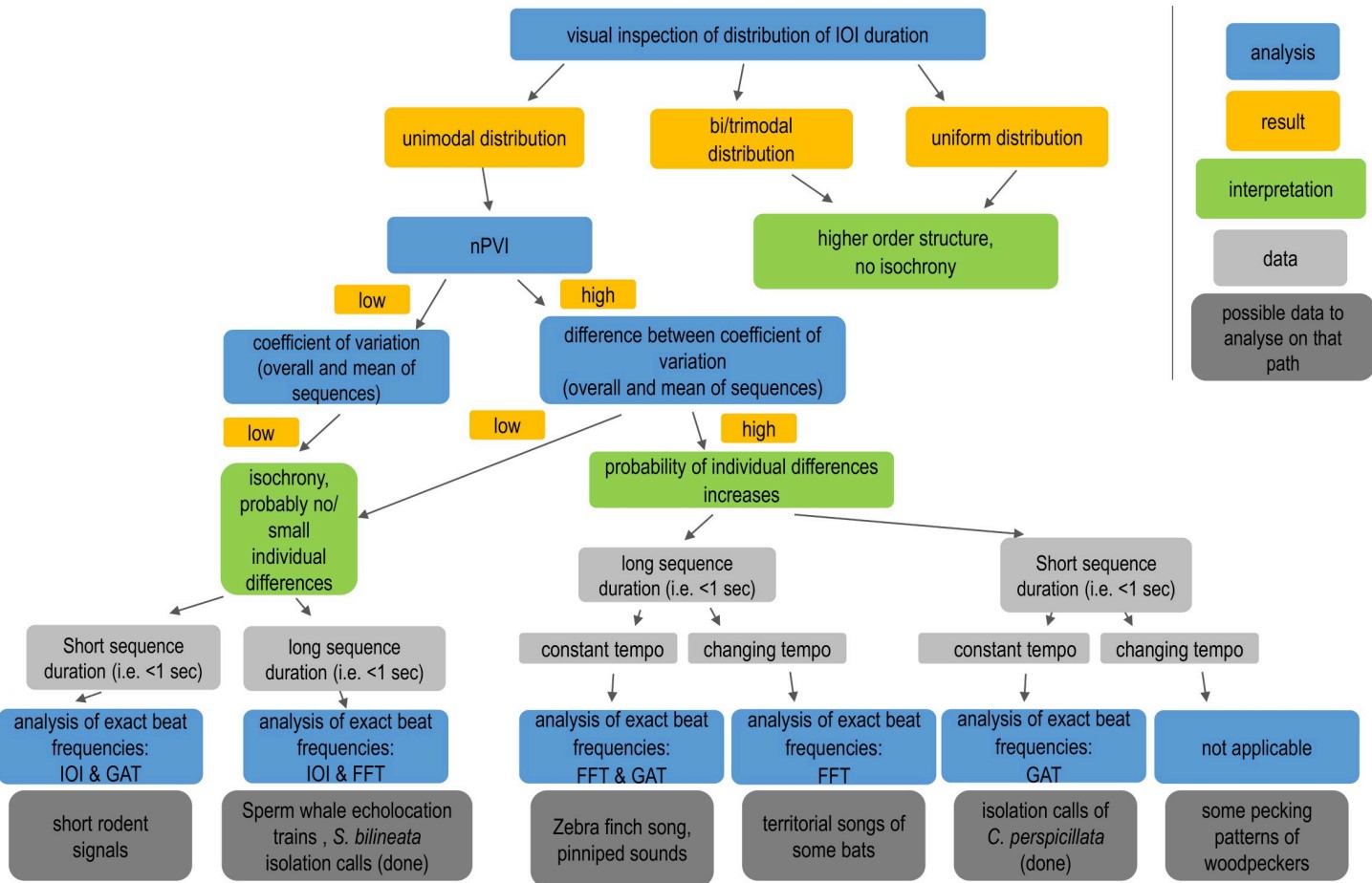

**Fig 7. Deciding on a method depending on the dataset and results.** The workflow starts with simple distributional measures such as IOI analysis and nPVI calculations. Questions to be answered in subsequent order are: 1) Is a dataset periodic? 2) If so, can we infer isochrony? 3) Assuming an isochronous pattern, how to analyze exact beat frequencies best, depending on the data at hand? The sequences we analyzed fall in three of the decision paths: *S. bilineata* isolation calls would be best to analyze with IOI or FFT; *C. perspicillata* would be best to analyze with the GAT approach, while *P. macrocephalus* echolocation click trains should be analyzed with IOI and FFT as well.

sampling rate; with the used sampling rate, frequencies of up to 100 Hz can be decomposed but this is not enough for faster signals (decomposition into frequencies up to half the sampling rate). Changing the sampling rate would on the other hand have implications on the frequency resolution. Duration of samples would need to be at least ~1 second for sampling frequencies up to 1000 Hz; if sequences are shorter, Fourier analysis is not suitable (Eq 7). On the other end of extremes, very slow signal sequences should not generate these kinds of problems. The frequency range to test for exact beat frequencies with the GAT approach would need careful consideration in this case though, because 2 Hz, which was the lower boundary in this case, might not be slow enough.

Looking at the different possible analysis paths we describe in the flowchart (Fig 7), we used data fitting into three different paths, leading to two different end categories. All three datasets show a unimodal distribution when looking at the IOI distribution. Results of nPVI and $C_V$ calculations differ. While echolocation click sequences of the sperm whale show low nPVI values and a small difference between $C_V$ (sequence) and $C_V$ (overall), both bat vocalizations do not fall into that category because nPVI values are higher. Nevertheless, since the difference between $C_V$ (overall) and $C_V$ (sequence) are also small in *S. bilineata*, we proceed in the

flowchart to the interpretation that, comparable to the sperm whale trains, isolation calls of *S. bilineata* are isochronous with probably no or only small individual differences. For *C. perspicillata* isolation calls, however, we conclude that even though isochronous, the probability for individual differences is increased, and therefore we proceed on a different path in the analysis.

Sequences of *S. bilineata* and *P. macrocephalus* are adequate in length (i.e mostly more than 1 second), therefore the frequency resolution in Fourier analysis is no problem and IOI analysis and Fourier analysis are most suitable for exact beat analysis. *C. perspicillata* sequences are shorter than 1 second and show a constant tempo, which would make the GAT approach the most suitable one. To give possible acoustic signal types for other paths, depending on the data might be from left to right in Fig 7: short call sequences of rodents, for example ultrasonic pulses of *Typhlomus chapensis* [42]; for sequences with a higher probability for individual differences that are above 1 second in duration and show a constant tempo one could think of male zebra finch song [5] or the vocalization sequences of pinnipeds such as the Northern elephant seal [6]. For a sequence with a changing tempo, one might think of a territorial song of some bat species that escalate and increase the tempo in the end [43–45]. Sequences, where none of the methods would be applicable, could, for example, be short, accelerating pecking patterns of woodpeckers [46].

The analysis of echolocation click trains of *P. macrocephalus* shows some interesting discrepancies between methods. Beat frequencies as known from the literature–often termed click rates or repetition rates in the respective literature–lie around 0.7–4 Hz [47, 48]. Using the IOI analysis, we get results fitting perfectly into that frame, which makes sense, as the same methodology is used. The other analyses also show way faster beat frequencies, even though not very prominently. The important message is that Fourier analysis and the GAT approach reproduce the overall pattern that most echolocation trains show beats as previously described in the literature. Nevertheless, it also shows the possibility of the oversimplification of IOI analysis; this needs more analyses but it might be possible that especially in more variable contexts than whale echolocation, IOI analysis is missing a lot of information, e.g. small differences that might be pronounced between individuals for discrimination purposes. It was already suggested that echolocation click beats of sperm whales may include this information [49].

There are a few general take home messages regarding methods to analyze the rhythm. Starting with the data, since all analyses rely on IOIs, elements need to be clearly separable and recordings need to have a good signal-to-noise ratio. Furthermore, the duration of a single sequence (i.e. duration between the first and the last element) should not be too short and a sequence should contain at the very least 3 elements for all methods to be applicable. In general, as many methods as possible should be applied to get a full picture of the data. Different methods have different flaws; by using various methods and comparing the results, artefacts or inconsistencies are easier to detect. Methods to calculate exact beat frequencies do have very different major flaws: Fourier analysis is not well applicable for very short sequences, because of the trade-of between time resolution in the original signal and frequency resolution in the Fourier signal (Eq 7). The GAT approach has issues with sequences changing in tempo since the optimization task is carried out for all elements within a sequence, such that one outlier can influence the results strongly. IOI analysis tends to oversimplify structures since it depends only on the mean of IOIs in a sequence, which is not depicting the variation in a sequence at all (Fig 5).

To enable reproducible rhythm analysis, one needs to provide at least the original IOI sequences of the data or even the raw acoustic signals with labels. Information on the generation of the binary sequence for the Fourier analysis is essential; this mainly refers to the time resolution used. If cluster analyses are run to detect individual patterns, reporting the used

distance measures, as well as clustering algorithm and distance threshold, are necessary to make results comparable between studies.

Considering all these things, rhythm analysis can be used to tackle many questions. Not only can we further investigate couplings of biological processes such as motor rhythms [50, 51], but it can be used to find possible guiding neural processes [5, 7] and can give valuable information for studies on the perception of temporal structures [52]. Especially in echolocating animals such as whales and bats, rhythm analysis yields a good background for studies on rhythm perception. Furthermore, rhythm analysis might prove to be a valuable tool for the analysis of vocal production learning, as was already suggested for example for the vocal learning in zebra finches, where very stereotyped elements are learned, with a difference only in the temporal structure [53]. In other species, one aspect of vocal production learning is for the animal to learn the temporal structure of an acoustic signal. Without knowing the beats produced by animal tutors and tutees, this is difficult to achieve [9].

Looking at a broad range of animal acoustic signals and uncovering broader patterns between animal taxa can, in the end, inform us about the origins and importance of periodicity and rhythmicity.

## Supporting information

**S1 Table. This document includes supporting information on artificial data and the validation and explanation of the goodness-of-fit value nGOF.**
(DOCX)

**S2 Table. This document includes supporting information on artificial data and expected and calculated results for the IOI analysis.**
(DOCX)

**S3 Table. This document includes supporting information on artificial data and the results for exact beat frequencies.**
(DOCX)

**S1 Fig. This document includes supporting information on artificial data and its results in form of a histogram panel.**
(DOCX)

**S2 Fig. This document includes supporting information on artificial data and the results of cluster analysis on their results as a figure.**
(DOCX)

**S1 Data. This datatable includes all Inter-Onset-Intervals of the original biological data that was analysed in this study.**
(XLSX)

## Acknowledgments

We thank the Dominica Sperm Whale Project and The Marine Bioacoustics Lab, Aarhus University, for sharing their sperm whale audio recording and helpful discussions, as well as the whole Knörnschild Lab for fruitful discussions.

## Author Contributions

**Conceptualization:** Lara S. Burchardt.

**Data curation:** Lara S. Burchardt, Mirjam Knörnschild.

**Formal analysis:** Lara S. Burchardt.

**Funding acquisition:** Lara S. Burchardt.

**Methodology:** Lara S. Burchardt.

**Software:** Lara S. Burchardt.

**Supervision:** Mirjam Knörnschild.

**Validation:** Lara S. Burchardt.

**Visualization:** Lara S. Burchardt.

**Writing – original draft:** Lara S. Burchardt.

**Writing – review & editing:** Lara S. Burchardt, Mirjam Knörnschild.

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
