## [Decision Letter · Decision Letter 0]

12 Jan 2020

Dear Dr Burchardt,

Thank you very much for submitting your manuscript, 'Method comparison for rhythm analysis of complex animal vocalizations', to PLOS Computational Biology. As with all papers submitted to the journal, yours was fully evaluated by the PLOS Computational Biology editorial team, and in this case, by independent peer reviewers. The reviewers appreciated the attention to an important topic but identified some aspects of the manuscript that should be improved.

We would therefore like to ask you to modify the manuscript according to the review recommendations before we can consider your manuscript for acceptance. Your revisions should address the specific points made by each reviewer and we encourage you to respond to particular issues Please note while forming your response, if your article is accepted, you may have the opportunity to make the peer review history publicly available. The record will include editor decision letters (with reviews) and your responses to reviewer comments. If eligible, we will contact you to opt in or out.raised.

- Supporting Information uploaded as separate files, titled 'Dataset', 'Figure', 'Table', 'Text', 'Protocol', 'Audio', or 'Video'.

We hope to receive your revised manuscript within the next 30 days. If you anticipate any delay in its return, we ask that you let us know the expected resubmission date by email at ploscompbiol@plos.org.

Sincerely,

Samuel J. Gershman

Deputy Editor

PLOS Computational Biology

[LINK]

Reviewer's Responses to Questions

**Comments to the Authors:**

Reviewer #1: In this manuscript, the authors showcase both old and new methods to test for the presence of temporal regularities in animal sounds. The ms is timely, well-written and sound, and it will be very helpful to a number of scientists. I was very happy to read this important piece of work.

I have a couple of major comments and several minor ones (see below).

1) Focus: The title, abstract and story of the ms are heavily centered on rhythm. This is great but the analysis methods and pipeline seem to me more about isochrony and periodicities rather than rhythm per se. A slight refocus (still keeping the rhythm, big picture in mind) may be good.

2) Baseline and validation: Some of the (mathematical) claims do not appear fully supported by the available text. I am sure the authors thought about this topic thoroughly, but if this is to become ‘the’ method for spotting/measuring temporal regularities in animal sounds, the maths/stats behind it need to be solid. Therefore, and practically, I’d encourage the authors to 1) make sure all the important claims and choices they make are fully supported by the maths and previous literature (see ‘detailed issues’ below for some specific points); 2) run the analyses also with some ‘synthetic’ temporal sequences, e.g. a perfectly isochronous one, a random sequence sampled from a uniform distribution, a random sequence sampled from a normal distribution or pink noise, etc.

Minor/detailed issues:

Title: Is ‘Method comparison’ correct in English?

line 14: an ‘and’ may be needed before ‘ecologists’

27: This sentence is a bit awkward.

37: What is the meaning of ’highly computational’? and can a FFT be considered such thing?

68-69 is this definition of animal sound broadly shared?

70: did the authors mean ‘connotations’?

120-121: Do the authors agree that rhythmic =/= periodic? In any case, how do they define periodic?

127-128: How do the authors define ‘beat’ here?

137: Why ‘IOI-‘ and not ‘IOI’?

174: Is ‘please’ necessary?

179-onwards: The ‘rhythm analyses’ section is a bit messy, mixing here and there definitions with explanations of concepts and actual methods/analyses/choices made in this paper.

187-188: Please check verbal tense agreement, here and elsewhere.

190: What is a ‘rhythm frequency’?

191: What is a ‘consistent rhythm’?

195-196: Is there an approximate/suggested cutoff value on n for using eq. (1) vs. (2)?

198: sizE

200: nPVI: Is this really the case? Why?

208: interval (only one ’l’)

209: A plot…is a possibility’?

234: add, e.g. ‘in the past’?

240: consider replacing with ’but is, instead, the mean’

249: Euler with capital ‘E’?

252-253: This statement should be proven mathematically.

260: what is the ‘respected…window’?

288-289: Is the Cv normally distributed? If not, how is it distributed across bouts?

Table 1: What is the meaning of the ‘crossed 0’ symbol? Isn’t it often used to denote the empty set?

325-327: This sentence is unclear. Also, and I am sorry if I am missing something, but I don’t see how the next sentence is linked and/or opposed to this sentence.

456: ‘to analyze’-> ‘analyzed’

460 and elsewhere, including the title: ‘periodicity and rhythmicality’ I believe this paper is rather about ‘periodicity and isochrony’, and rhythm plays a more indirect role

462: Doesn’t GOF need validation?

470: The main verb is missing.

503: “probability…increased” has a strong, quantitative connotation, which is at present not established in the manuscript.

552-554: I believe mentioning Julia Hyland Bruno’s work here would be quite relevant.

Figure 1: I wonder whether the authors should reference here the Frontiers paper by Ravagnani Bowling and Fitch containing a similar schematic.

Figure 7: Why is the cutoff at exactly 1 sec?

References: There are a few issues/typos, please check them. For instance, ref #14: Celma & Toro were authors not editors. Also, a paper describing the work in that abstract recently appeared in J of Comparative Psychology.

Reviewer #2: Review of manuscript PCOMPBIOL-D-19-02059, ‘Method comparison for rhythm analysis of complex animal vocalizations’.

This manuscript focuses on an important subject in computational biology, and the authors are experienced in the topic. However, I find that the presentation is a bit too focused on the several selected species, and this concentration will prevent the paper from attaining its stated goal : showing biologists – who may not have considerable experience with acoustic signals and rhythm – how they might best conduct their analyses.

Main points

1 It would be more helpful to begin with an overview of rhythm in acoustic signals and introduce the key features that can be identified among many animal species. Some attention should be paid to ‘nested’ parameters ; e.g. calls that are comprised of syllables, which in turn are comprised of pulses, each one of these units having a rhythm, unit duration, and inter-unit gap (or onset-to-onset interval). Ideally, this framework would be established without reference to the specific terms (e.g. syllables, pulses), as different authors tend to use their own definitions.

2 The authors note the potential importance of acoustic rhythm in biology, but it would be helpful to specify the different fields of biology where measurement of rhythm could be valuable and what aspects might be most important to measure. For example, rhythm could be critical for assessing species identity, gender or developmental stage, an individual’s social status, and individual identity. These several factors might demand different types of analysis in order to extract the most relevant information.

3 More attention should be paid to using terms in a more rigorous way and to being consistent with their use in other fields of science. For example, ‘beat’ has a very specific meaning in physics (beat frequency = difference between two different frequencies), whereas the authors are using the term in a loose manner to refer to animal calls. While ‘beat’ may have been used in this loose fashion by workers studying a particular group of animals, it is not at all general and will cause confusion. Many other examples exist in the manuscript. Another issue is the relationship between rhythm and periodicity, wherein the authors assume that rhythms always entail a certain period of repetition of the relevant units. However, workers studying human language often recognize a ‘language rhythm’ that does not depend on any particular period. This issue needs careful consideration, particularly when dealing with mammalian vocalizations.

4 The manuscript is fraught with an excessive amount of jargon, a feature that is likely to deter the generalist reader interested in applying rhythm analysis for the first time. Some jargon is, of course, necessary, but these specialized terms begin very early in the manuscript and dominate it.

5 Using examples of actual vocalizations is helpful, but they should be presented only to show what sorts of analyses are applicable in particular cases. However, the authors have built their manuscript around several vocalization examples, and the reader is not left with a general idea of where to use method a, method b, etc. And, a given method may reveal certain critical features in vocalization a while another method reveals other features in the same vocalization ; i.e. the application of several methods may be best. More importantly, the several vocalization examples presented by the authors represent a very small portion of the range of vocalization rhythms that biologists can encounter.

6 Studying acoustic rhythm is a fundamental aspect of animal behavior and physiology, but animals often exhibit their rhythms in the company of conspecifics and thereby generate rhythm interactions. Some mention / treatment of measuring rhythm interaction would be helpful.

Minor point

7 A simple method that often works to identify rhythm periodicity is harmonic analysis : An individual may call with a given free-running (endogenous) rhythm period, but owing to various internal and external stimuli he/she can miss one or several calls at times. By arranging the individual’s inter-call intervals in time bins, periodicity is revealed by the presence of most inter-call intervals in bin X, a smaller number of intervals in bin 2X, a yet smaller number in bin 3X, etc.

**Have all data underlying the figures and results presented in the manuscript been provided?**

Reviewer #1: Yes

Reviewer #2: Yes

PLOS authors have the option to publish the peer review history of their article (what does this mean?). If published, this will include your full peer review and any attached files.

Reviewer #1: No

Reviewer #2: No

---

## [Editor Report · Decision Letter 1]

28 Feb 2020

Dear Ms. Burchardt,

We are pleased to inform you that your manuscript 'Comparison of methods for rhythm analysis of complex animals’ acoustic signals' has been provisionally accepted for publication in PLOS Computational Biology.

Best regards,

Samuel J. Gershman

Deputy Editor

PLOS Computational Biology

---

## [Editor Report · Acceptance letter]

23 Mar 2020

PCOMPBIOL-D-19-02059R1 

Comparison of methods for rhythm analysis of complex animals’ acoustic signals

Dear Dr Burchardt,

I am pleased to inform you that your manuscript has been formally accepted for publication in PLOS Computational Biology. Your manuscript is now with our production department and you will be notified of the publication date in due course.

With kind regards,

Bailey Hanna
